

# Identification of novel genes associated with litter size of indigenous sheep population in Xinjiang, China using specific-locus amplified fragment sequencing technology

Haiyu Ma[1], Chao Fang[2], Lingling Liu[1], Qiong Wang[1], Jueken Aniwashi[1], Yiming Sulaiman[1], Kezierkailedi Abudilaheman[3] and Wujun Liu[1]

[1] College of Animal Science, Xinjiang Agriculture University, Urumqi, Xinjiang, China
[2] Department of Veterinary Managment of Animal Resources, Faculty of Veterinary Medicine, University of Liège, Liège, Belgium
[3] People's Congress of Xinjiang Uygur Autonomous Region, Urumqi, Xinjiang, China

Corresponding author
Wujun Liu, lwj_ws@163.com

## ABSTRACT

**Background**. There are abundant sheep breed resources in the Xinjiang region of China attributing to its diverse ecological system, which include several high-litter size sheep populations. Previous studies have confirmed that the major high prolificacy gene cannot be used to detect high litter size. Our research team found a resource group in Pishan County, southern Xinjiang. It showed high fertility with an average litter size of two to four in one birth, excellent breast development, and a high survival rate of lambs. In the present study, we used this resource as an ideal sample for studying the genetic mechanisms of high prolificacy in sheep.

**Methods**. Indigenous sheep populations from Xinjiang, with different litter sizes, were selected for the research, and specific-locus amplified fragment sequencing (SLAF-seq) technology was used to comprehensively screen single nucleotide polymorphisms (SNPs) from the whole genome that may cause differences in litter size. Novel genes associated with litter size of sheep were detected using genome-wide association studies (GWAS), providing new clues revealing the regulation mechanism of sheep fecundity. Candidate genes related to ovulation and litter size were selected for verification using Kompetitive Allele Specific polymerase chain reaction (KASP) cluster analysis.

**Results**. We identified 685,300 SNPs using the SLAF-seq technique for subsequent genome-wide analysis. Subsequently, 155 SNPs were detected at the genome-wide level. Fourteen genes related to sheep reproduction were notated: *COIL, SLK, FSHR, Plxna3, Ddx24, CXCL12, Pla2g7, ATP5F1A, KERA, GUCY1A1, LOC101107541, LOC101107119, LOC101107809*, and *BRAF*. Based on literature reports, 30 loci of seven genes and candidate genes (*CXCL12, FSHR, SLK, GUCY1A1, COIL, LOC101107541*, and *LOC101107119*) related to ovulation and litter size were selected for verification using KASP cluster analysis. Among them, nine loci of three genes were successfully genotyped. Three loci of *FSHR* (GenBank ID: 443299, g. 75320741G>A site), *GUCY1A1* (GenBank ID: 101110000, g. 43266624C>T site), and *COIL* (GenBank ID: 101123134, g. 7321466C>G site) were found to be significantly or extremely significantly associated

with litter size. These three loci are expected to be used as molecular markers to determine differences in litter size in sheep.

## INTRODUCTION

Fertility is one of the most important economic traits in sheep. Sheep populations with high reproductive performance show two to three times higher production efficiency and economic benefit from lambs than those with low reproductive performance do. Therefore, detection of molecular markers of high fecundity in sheep is of great significance in revealing the genetic basis of sheep reproductive traits, improving sheep breeds by molecular breeding, and establishing core groups or breeding new lines. Therefore, reproductive traits have become a research hotspot in sheep breeding.

Among sheep breeds worldwide, few exhibit high litter size, early sexual maturity, and perennial oestrus. Presently, a few major genes affecting high fecundity have been found in Cambridge and Belclare (*Hanrahan et al., 2004*), Icelandic (*Eiriksson & Sigurdsson, 2017*), Romanov (*Deniskova et al., 2017*), Finnish (*Mullen & Hanrahan, 2014*), and other high fecundity sheep breeds abroad, whereas domestic studies have mainly focused on Hu (*Yue, 1996*) and Small Tail Han sheep (*He et al., 2012*). The results show that different major genes affect the litter size of various sheep breeds. As a result, the major genes identified in small-tailed Han sheep have been selected as markers for domestic mutation detection for sheep litter size, which has led to pseudoscience in this field.

Xinjiang has a diverse ecological system with abundant sheep resources, including several high litter size sheep populations. Previous studies have confirmed the existence of the high prolificacy major gene *FecB* mutation in the Cele black sheep (*Jiang et al., 2017*) and Duoliang sheep populations (*Wang et al., 2017*), but it cannot be used to detect high litter size. Our research team found a resource group in Pishan County, southern Xinjiang, for the indigenous prolificacy sheep breed in Xinjiang. The ewes are oestrus all year round mainly because of natural mating and, therefore, they are bred throughout the year. This breed showed high fertility with an average litter size of two to four lambs per birth, excellent breast development, and a high survival rate of lambs. Bashbai sheep are single-breasted, seasonally oestrus sheep found in the Tacheng area of Xinjiang. Breeding is mainly via artificial insemination, and the oestrus period is typically in November. The lambing rate is 103%, milk yield is stable, and lamb survival rate is 98%. These breeds is considerably different in the number of lambs produced. Consequently, we considered them an ideal sample for studying the genetic mechanisms of high prolificacy in sheep.

In this study, ewes with different litter size (one to four) in Xinjiang were selected as research materials. Genome-wide association studies (GWAS) based on the specific-locus amplified fragment sequencing (SLAF-seq) technique were used to identify single nucleotide polymorphisms (SNPs) that might cause differences in litter size.

## MATERIALS & METHODS

### Sample collection and DNA extraction

A total of 126 sheep from two populations were used in this study, including 62 Hetian sheep from Pishan county of Hetian city of Xinjiang (East longitude 77°31′–79°38′, latitude 35°22′–39°01′), and 64 Bashbay sheep from Yumin county of Tacheng state of Xinjiang (East longitude 82°12′–83°30′, latitude 45°24′–46°3′). Whole blood samples (6 mL) were collected from the jugular veins of the sheep and transferred into ethylenediaminetetraacetic acid (EDTA) anticoagulant tubes. Genomic DNA was extracted using a standard phenol chloroform extraction method (*Köchl, Niederstätter & Parson, 2005*) for subsequent experiments. The study design was approved by the appropriate ethics review board. The University of Xinjiang Agricultural University approved the use of its facilities for the study (Animal protocol number: 2017010).

### Construction of SLAF-seq Library and High-throughput Sequencing

The current sheep genome (Oar_v4.0) was selected as the reference genome to simulate the restriction enzyme digestion and identify the expected SLAF yield. Hpy166II + EcoRV-HF enzyme was selected for enzymatic digestion (*Davey et al., 2013*). Furthermore, to evaluate the accuracy of the digestion strategy, *Oryza sativa indica* ( http://rapdb. dna. affrc. go. JP/) was selected (*Li et al., 2009b*). Subsequently, data of 126 individuals were used to construct the SLAF library (*Kozich et al., 2013*) and for sequencing after quality inspection. To evaluate the accuracy of the enzyme test, Nippon Sunshine was selected as the control for sequencing.

### Analysis of SLAF-seq data and identification of SNP loci

According to SLAF tags, the SNP locus information was analysed and screening criteria was set at minor allele frequency (MAF) >0.05. The development of SNP markers was based on the sheep reference genome using BWA (*Li & Durbin, 2009*) to compare the sequenced reads to the reference genome, and GATK (*Mckenna et al., 2010*) and SAMtools (*Li et al., 2009a*) to SNP calling. The intersection of the SNP markers obtained using the two methods was used as the final reliable data set of SNP markers. Sequencing reads of the control were compared with their reference genomes using SOAP software. The double-end contrast efficiency was 92.46%, and the contrast efficiency was normal. The enzymatic cleavage efficiency of the control was 93.60%, indicating that the enzymatic cleavage reaction was normal.

### Population structure

Principal component analysis (PCA) was performed using PLINK1.9 (*Purcell et al., 2007*), and the ggplot2 package in R (v3.4.4) was used to generate the PCA figure (*Wickham, 2015*). We removed the SNPs in linkage disequilibrium in PLINK 1.9 using the command (–indep-pairwise 50 5 0.2).

### GWAS

Based on SNP analyses, the general linear model (GLM) and mixed linear model (MLM) of the TASSEL (*Bradbury et al., 2007*) software (http://www.maizegenetics.net) were used

to obtain the correlation values. The Q matrix of the sample population structure was calculated using Admixture software (*Alexander, Novembre & Lange, 2009*) and the K matrix of the relationship between samples was calculated using SPAGeDi software (*Hardy & Vekemans, 2002*). The GLM uses the group structure information, whereas the MML uses the information of the population structure and the kinship relationship. The fixed effects of the MLM model are parity and population, the random effects are related, and each SNP site is finally associated with a value. Manhattan chart analysis and Q-Q Plot were used to analyse the population structure and their diagrams were both drawn using R language (*Zhiwu et al., 2010*).

## Bioinformatics analyses

We combined several commonly used bioinformatics databases, such as the National Center for Biotechnology Information (NCBI, http://www.ncbi.nlm.nih.gov/), University of California Santa Cruz (UCSC, http://genome.ucsc.edu/), and Ensembl (http://asia.ensembl.org/index.html) to locate significant SNPs.

The position of the significant SNPs was located, and for SNP markers that were not within the gene, candidate genes 500 kb up-stream and down-stream of the significant SNP site were searched to determine the linkage disequilibrium between the markers. Candidate gene functions were identified and analysed using the online gene enrichment software DAVID 6.7 (https://david.ncifcrf.gov/home.jsp). Venny2.1.0 (http://bioinfogp.cnb.csic.es/tools/venny/index.html) was used to draw a Venn diagram of the overlap sites of the two methods.

## KASP typing verification

KASP typing was performed at 30 loci of the seven candidate differential genes (*FSHR, COIL, GUCY1A1, CXCL12, SLK, LOC101107541,* and *LOC101107119*), which were annotated. Nine pairs of successfully typed primers are listed in Table 1. A one-way analysis of variance (ANOVA) and $t$-test were performed using the statistical package for the social sciences (SPSS) version 19.0 software to analyse the association between litter size and mutant locus genotypes.

## RESULTS

### Establishment of database and sequencing evaluation

To obtain the actual SLAF tags used in this study, 62 and 64 Hetian and Bashbai sheep, respectively were subjected to SLAF-seq using the same enzyme combinations as those used in computer restriction analysis. A total of 854.88 Mb reads data were obtained for all individuals, and the average Q30 and GC contents were 91.74% and 42.14%, respectively, indicating that the SLAF-seq process was normal and available. After genome comparison and SNP mining, 5,218,278 population SNPs were found using all individuals. The completeness was 0.5 times, and the genomic frequency was 0.05 filtered to 685,300 SNPs sites, and 685,300 SNPs were identified for subsequent analysis.

**Table 1  Information of four genes.**

| Gene | Position | Primer Sequence |
|------|----------|-----------------|
| GUCY1A1 | 43266624 | F1: GAAGGTCGGAGTCAACGGATTGGAGTGGGCCAGCAGCTAC |
|  |  | F2: GAAGGTGACCAAGTTCATGCTGGAGTGGGCCAGCAGCTAT |
|  |  | R1: GTTCTTGTCAGGGACACCTGG |
| SLK | 23608558 | F1:GAAGGTCGGAGTCAACGGATTCTTGCGAGATGAAGCCAAGC |
|  |  | F2:GAAGGTGACCAAGTTCATGCTCTTGCGAGATGAAGCCAAGT |
|  |  | R1: ACATTCTGAAATTTGGACAGCTC |
| COIL | 7314134 | F1: GAAGGTCGGAGTCAACGGATTCCATGAAAGAACCTGGGAAA |
|  |  | F2: GAAGGTGACCAAGTTCATGCTCCATGAAAGAACCTGGGAAC |
|  |  | R1: CCTCAGCTCCATTTTCGTTG |
| COIL | 7321466 | F1: GAAGGTCGGAGTCAACGGATTGACTCCGAGGAGGAATCGC |
|  |  | F2: GAAGGTGACCAAGTTCATGCTGACTCCGAGGAGGAATCGG |
|  |  | R1: GTGGCATGGTCGTCCGTAC |
| COIL | 7321563 | F1:GAAGGTCGGAGTCAACGGATTGCACAGTCTGTGAAAGAGTGGA |
|  |  | F2: GAAGGTGACCAAGTTCATGCTGCACAGTCTGTGAAAGAGTGGG |
|  |  | R1: TCTAGCAGGAAGAGCTTTAGGG |
| FSHR | 75132817 | F1: GAAGGTCGGAGTCAACGGATTAGCCCAAGCTCAGGAATGC |
|  |  | F2: GAAGGTGACCAAGTTCATGCTGAGCCCAAGCTCAGGAATGT |
|  |  | R1: GGTGGATGGATAAGTAAACATGG |
| FSHR | 75320579 | F1: GAAGGTCGGAGTCAACGGATTGGACAGGGAAGACTCACTCACA |
|  |  | F2: GAAGGTGACCAAGTTCATGCTGACAGGGAAGACTCACTCACG |
|  |  | R1: CTCACCTACCCCAGCCACT |
| FSHR | 75320741 | F1: GAAGGTCGGAGTCAACGGATTGATATTTCAAGAACCAGGATCCA |
|  |  | F2: GAAGGTGACCAAGTTCATGCTATATTTCAAGAACCAGGATCCG |
|  |  | R1: CAGCTTCTTAAGATTTTCTAAGCC |
| FSHR | 75132820 | F1: GAAGGTCGGAGTCAACGGATTATGATGCTGGCAGCATGGT |
|  |  | F2: GAAGGTGACCAAGTTCATGCTATGATGCTGGCAGCATGGC |
|  |  | R1: CATCACCCACGCCATGCAG |

## Population stratification assessment

The results of the PCA (Fig. 1) show that PC1 and PC2 had variances of 3.02% and 2.15%, respectively. Furthermore, the results showed that Hetian and Bashbay sheep were separated by PC1 and there was no mixing between the two populations, which provided a foundation for the subsequent GWAS.

## GWAS

In this study, GLM and MLM were used to analyse the GWA of litter size traits in Hetian and Bashbai sheep. A Bonferroni correction of $\alpha = 0.1\%$ was applied for genome-wise thresholds (significance threshold $= -\log_{10} [\alpha/$number of independent SNPs$]$). SNPs with $p$-values below 1.459e$-07$ (0.1/685300) were considered significantly associated with the phenotype. GWAS results showed that 111 and 44 SNPs were significantly associated with the genome in the GLM and MLM, respectively, and 155 significant loci were identified. Manhattan charts of litter size traits are shown in Figs. 2 and 3. A total of 25 SNPs were

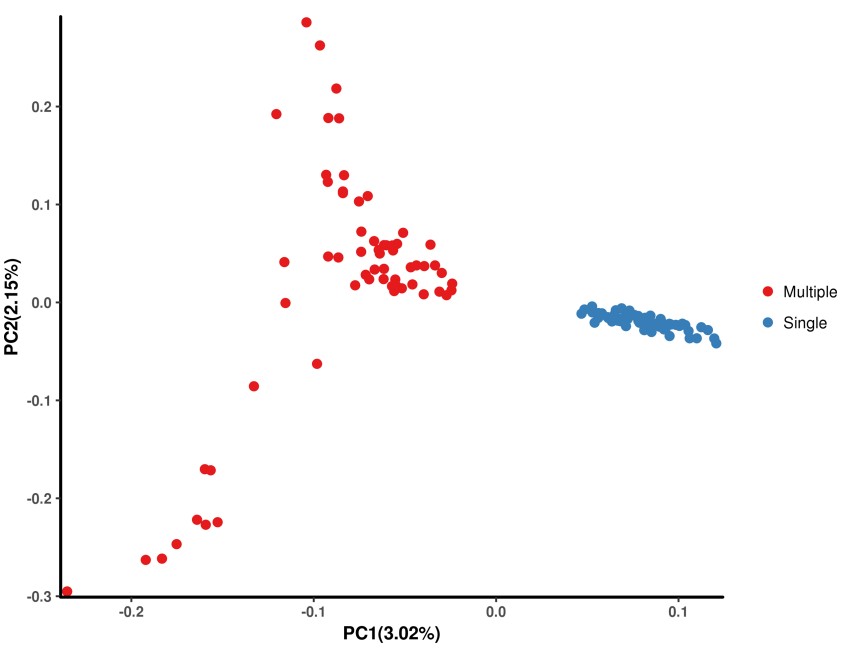

**Figure 1** **Principal component analysis (PCA) of Hetian (Mulitiple) and Bashbay (Single) sheep breeds.**

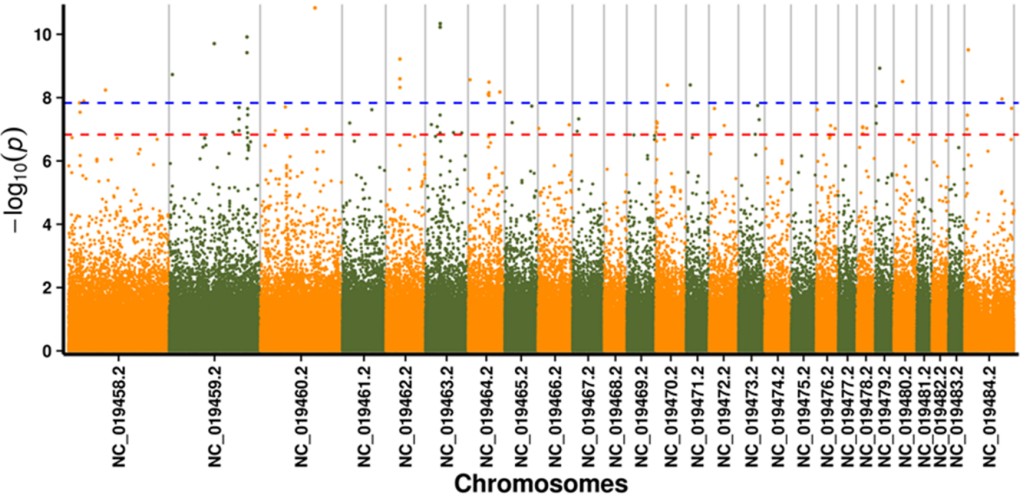

**Figure 2** **Manhattan plot for genome-wide association study on GLM model.** Note: the scale on the *X*-axis represents ID of chromosomes, NC_019458.2–NC_019483.2, represents ID 1-26 of chromosomes, and the *X* chromosome is represented by NC_019484.2. The scale on the *Y*-axis is the −log10 (*P*-value) score of association analysis. The red dashed line indicates genome-wide significance of suggestive associa-tion.

detected using both methods (Fig. 4). The QQ-plot diagram (Figs. 5 and 6) shows a large deviation in the SNP point, which at this site was considered to be caused by the genetic effect of this SNP mutation.
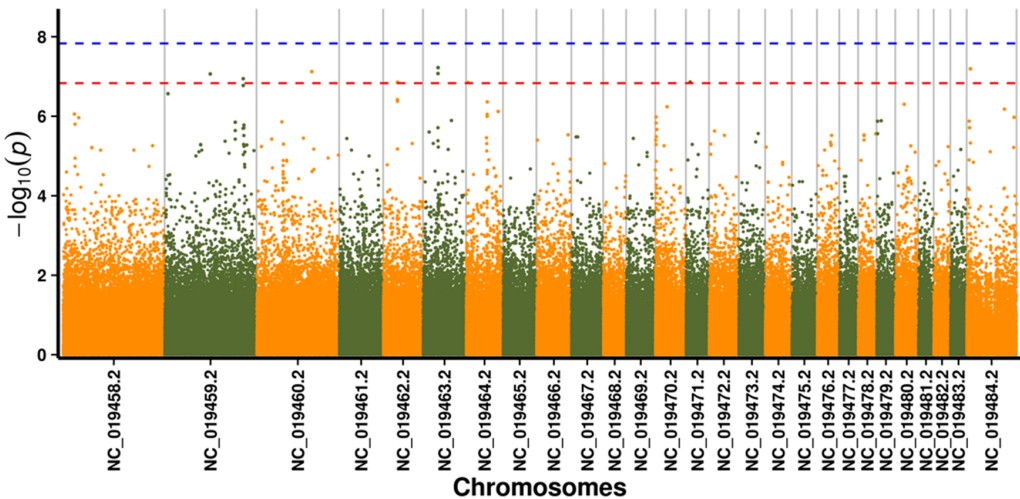

**Figure 3   Manhattan plot for genome-wide association study on MLM model.** Note: the scale on the X-axis represents ID of chromosomes, NC_019458.2–NC_019483.2, represents ID 1-26 of chromosomes, and the *X* chromosome is represented by NC_019484.2. The scale on the *Y*-axis is the −log10 (*P*-value) score of association analysis. The red dashed line indicates genome-wide significance of suggestive association.

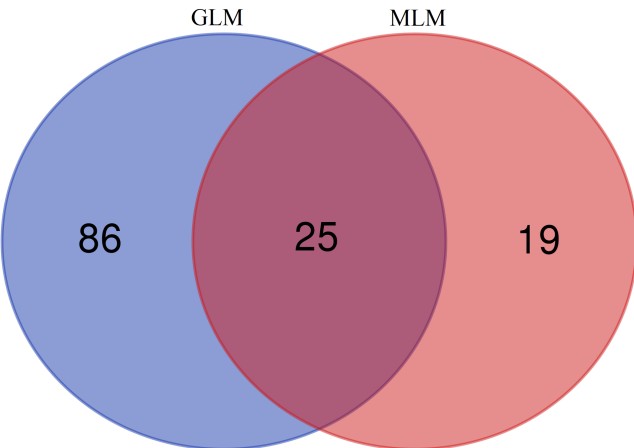

**Figure 4   Venn diagram of two methods.**

## Gene function annotation

Using Oar_v4.0 sequence information of sheep genome and common database information such as that from NCBI, SNP loci with significant GWAS results were analysed and annotated. A total of 133 genes were annotated in the two models, including *FRS2, RGS3, MDH1, IMPA1*, and *KCNE3*, which are involved in the differentiation and survival of nerve cells. Some of them are new genes that have not been clearly labelled, and their functions need to be further studied; 14 of these genes were related to reproduction. *COIL, SLK, Plxna3*, and *Ddx24* genes affect the development of ovaries and follicles in sheep; *CXCL12,*

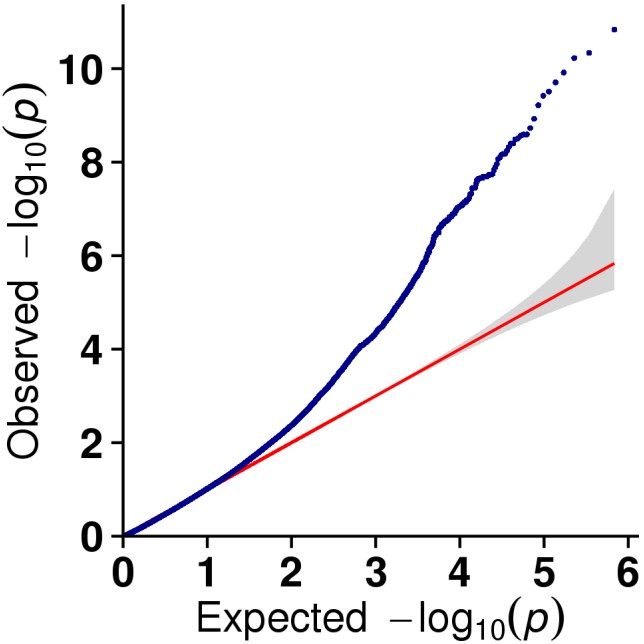

**Figure 5** The results of quantile-quantile (Q-Q) plot for litter size trait in the GLM model.

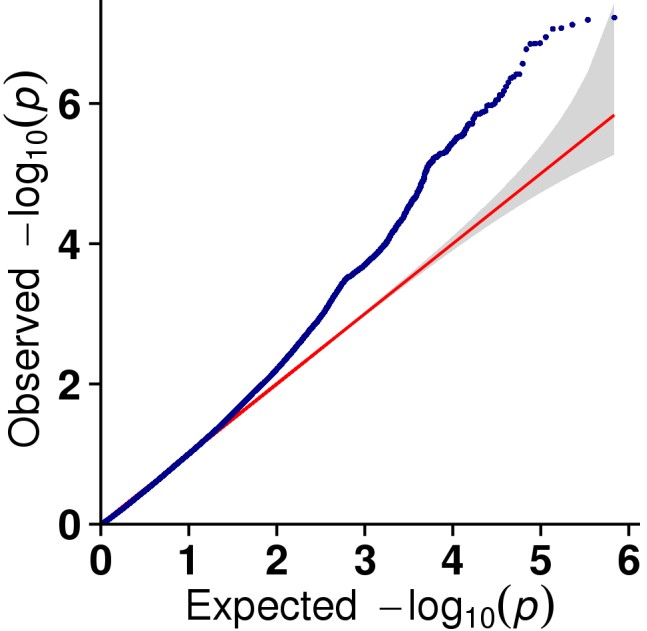

**Figure 6** The results of quantile-quantile (Q-Q) plot for litter size trait in the MLM model.

*Pla2g7, ATP5F1A,* and *KERA* affect the early development of the placenta and placental membrane; *GUCY1A1, LOC101107541, LOC101107119,* and *LOC101107809* participate in the oxytocin signalling pathway and indirectly regulate the production of ovarian steroids; and *BRAF* and *Ddx24* are the most expressed in the uterus.

### Enrichment analysis results

Genecards and DAVID6.7 online websites were used to analyse the functions of the candidate genes while Gene Ontology (GO) and Kyoto Encyclopedia of Genes and Genomes (KEGG) online websites were searched for candidate genes. Enrichment analysis revealed that these differential genes are mainly involved in pathways such as oxytocin signalling, amino acid biosynthesis, neurotrophic signalling, pentose phosphate, and Wnt signalling. In addition to the *FSHR, SLK, GUCY1A1,* and *LOC1001107541* genes, a few others are involved in the oxytocin signalling pathway, which regulates the litter size of sheep by regulating follicle stimulating hormone receptors, and ultimately forms a multi-foetal sheep population.

### Functional validation of genes related to litter size regulation

Based on the results of previous studies and comprehensive analysis of gene function, 30 missense mutation sites of seven genes (*CXCL12, FSHR, SLK, GUCY1A1, Coil, LOC101107541,* and *LOC101107119*) closely related to ovulation and litter size were screened, and 126 individuals with different litter size were typed using KASP. Nine sites of the final three genes (*FSHR, GUCY1A1,* and *COIL*) were successfully typed. Only one genotype was identified at 21 loci of the remaining four genes, which could not be used to determine the genotype.

### Association analysis between different genotypes and litter size

The association analysis between successfully typed genotypes and individuals with different litter size was showed significant differences or significant correlations in litter size among different genotypes of *FSHR* (g.75320741G>A), *GUCY1A1* (g.43266624C>T), and *COIL* genes (g.7321466C>G, Figs. 7, 8 and 9).

At the g.75320741 locus of the *FSHR* gene, the average litter size of G/G genotype individuals was significantly higher than that of A/A genotype individuals was ($P = 0.004$). At the g.43266624 locus of the *GUCY1A1* gene, the average litter size of C/C genotype individuals was significantly higher than that of T/T genotype individuals was ($P = 0.038$). At g.7321466 locus of COIL gene, the average litter size of C/C genotype individuals was significantly higher than that of T/T genotype individuals ($P = 0.042$, Fig. 10).

## DISCUSSION

### Evaluation of reliability of SLAF-seq technology

In this study, the SLAF-seq method used to identify SNPs located in the genome of Chinese indigenous sheep populations detected >685,300 SNPs. In addition, 133 genes were annotated by comparing and analysing the loci of SNPs with significant GWAS results for reproductive traits.

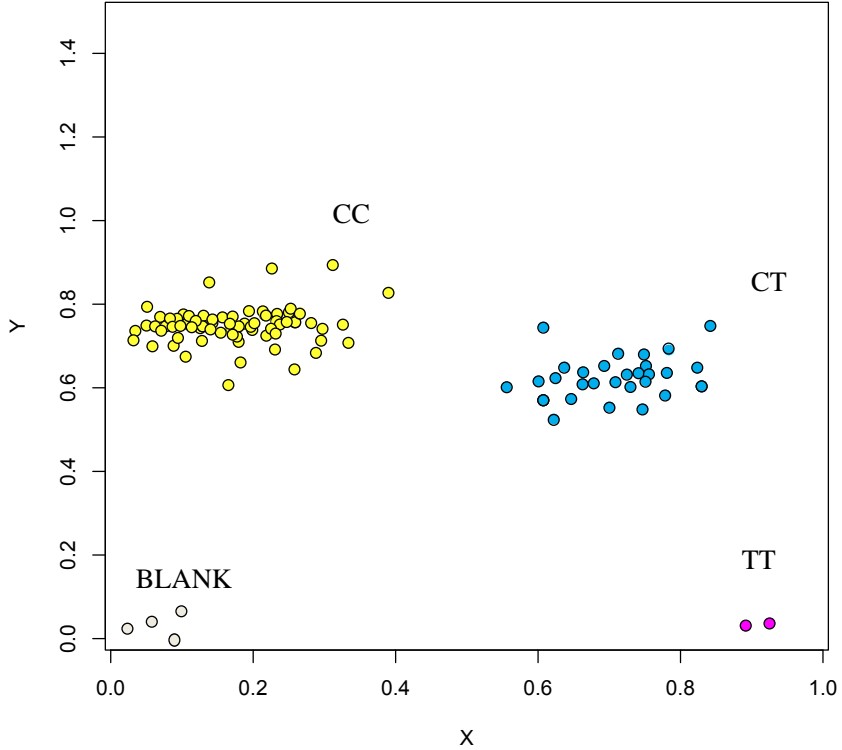

**Figure 7** **Genotyping results of the GUCY1A1 gene.**

SLAF-seq is a simplified deep genome-sequencing technique, which can be used to obtain and accurately type numerous molecular markers using bioinformatics. This technology has been successfully implemented in studying many species such as cotton (*Li et al., 2017*) (IF: 7.44) and soybean (*Han et al., 2016*) (IF: 7.33). In this study, 1192168 SLAF tags from the sheep reference genome were predicted using the Hpy166II + EcoRV-HF® enzyme combination. The average sequencing depth was 13.06x, and 685,300 SNPs were found. Therefore, the SLAF-seq method used in this experiment provided comprehensive genomic variation information. Considering the enormous differences in genome sequences between European and Chinese sheep breeds, these experimental data show that SLAF-seq is a powerful method with considerable potential for use in studying more breeds. Therefore, the SLAF-seq method could be considered a highly efficient option for sheep genome research.

## Molecular markers and candidate genes of litter size in sheep

Genetic markers for the number of lambs in sheep have been studied, but the reports are limited. More studies have focused on the ovulation rate than on the litter size. Current studies have shown that *GDF9, BMP15*, and *BMPR-IB* genes and 13 mutation loci (*FecBB, FecXB, FecXG, FecXGR, FecXH, FecXI, FecXL, FecXO, FecXR, FecGH, FecGT,* and *V371M*) (*Mullen & Hanrahan, 2014*) are the major genes affecting litter size or ovulation rate in sheep. However, these markers are not stable in other sheep breeds and are not the major
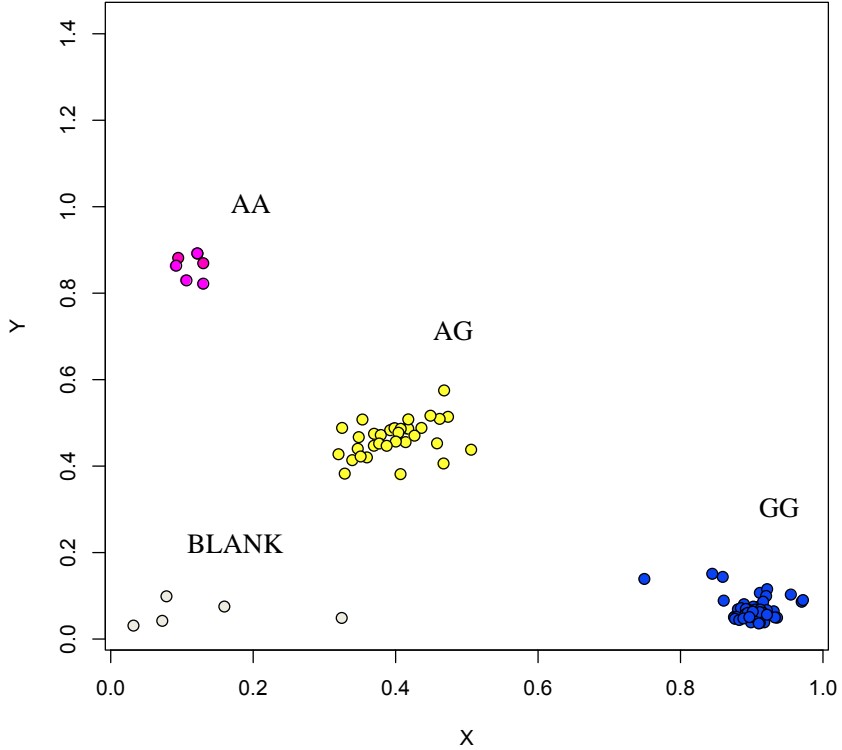

**Figure 8** **Genotyping results of the FSHR gene.**

genes affecting their litter size, indicating that there are other genes responsible for litter size and interspecies differences. High fecundity of sheep breeds is very rare worldwide. Xinjiang has very rich sheep resources and the Hetian sheep breeds used in this experiment, which often produce many lambs, have rarely been used an experimental model.

In this study, 30 loci of seven ovulation-related genes (*FSHR, COIL, GUCY1A1, CXCL12, SLK, LOC101107541,* and *LOC101107119*) were selected for KASP validation. Results showed that nine out of the 30 loci of three genes (*CXCL12, FSHR,* and *COIL*) were successfully genotyped. The other four genes are rarely reported in livestock and have mostly been studied in humans and mice. Among them, the *SLK* gene affects the development of the ovary and follicle (*An, 2012*), *CXCL12* affects early development of the placenta and placental membrane (*Quinn et al., 2016*; *Sanchez et al., 2017*), and *LOC101107541* and *LOC101107119* participate in the oxytocin signalling pathway and indirectly regulate the production of ovarian steroids. The results need to be further verified.

## Relationship between *FSHR* gene and sheep reproduction

This study concluded that the *FSHR* gene may be one of the important genes affecting litter size of Hetian sheep. Follicular stimulating hormone receptor (FSHR) is a member of the glycoprotein superfamily of G protein-coupled receptors and plays an important role in follicular development in animals. FSHR is mainly expressed in granulosa cells of follicles in super ovulated and normal lambs, and there are positive signals in primordial

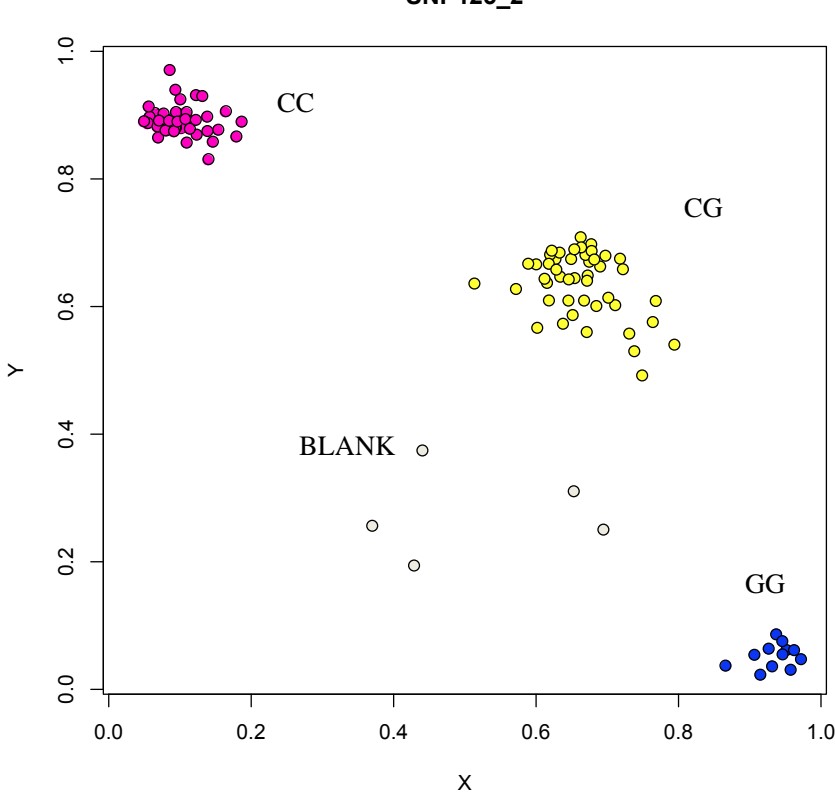

**Figure 9** **Genotyping results of the COIL gene.**

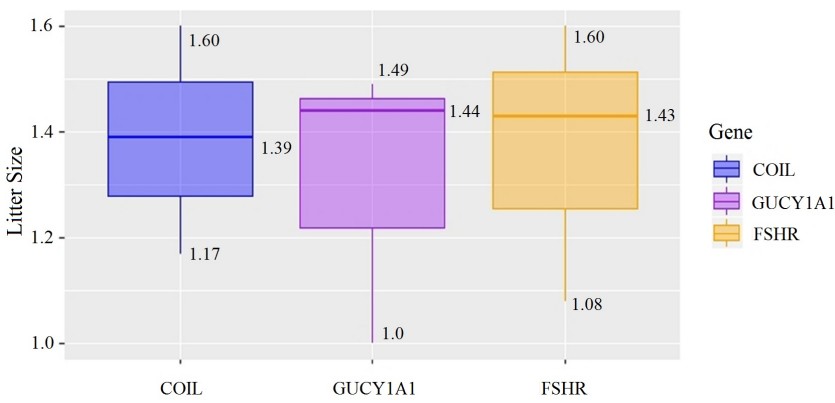

**Figure 10** **Correlation analysis between different genes and litter size.**

follicles. No positive signals are observed in primordial follicles in normal adult sheep, and the expression of FSHR is decreased in large dominant follicles (*Chu et al., 2012*). Some studies have reported the full-length coding sequences of six *FSHR* spliced forms in sheep. The open reading frames are 695aa, 694aa, 648aa, 633aa, 595aa, and 533aa. While 533aa

is not found in the ovaries of lambs, 694aa and 648aa are not found in adult sheep (*Jiang, 2014*). In this study, *FSHR* was found to be associated with follicular development.

*Chu et al. (2012)* detected 50 SNPs in the regulatory region of the *FSHR* gene in two high- reproductive (small-tailed Han and Hu) and two low-reproductive (Kaolidai and Chinese Merino) sheep breeds. Four mutations in the sheep *FSHR* gene were detected using polymerase chain reaction-single-strand conformation polymorphism (PCR-SSCP) technology, suggesting that the *FSHR* gene may significantly affect the litter size of sheep (*Chu et al., 2012*). *Pan et al. (2014)* cloned 50 flanking regions of the sheep *FSHR* gene and analysed its genomic structure. Reverse transcription-quantitative PCR (RT-qPCR) showed that *FSHR* was widely expressed in the analysed sheep tissue. In addition, the homologous mutation of the *FSHR* gene was found to be significantly correlated ($P < 0.1$) with litter size (*Pan et al., 2014*). Some of the present observations are consistent with those of previous studies.

In this study, the g.75320741G>A locus of the *FSHR* gene was successfully classified into three genotypes, namely GG, GA, and AA in the Xinjiang Hetian and Bashbai sheep breeds. Correlation analysis comparing the different genotypes and litter size showed that the GG genotype was highly correlated with litter size, which further verified the reliability of SLAF-seq and verified that *FSHR* is a candidate gene affecting the litter size in sheep.

## Relationship between Coilin (*COIL*) gene and sheep reproduction

Coilin is a characteristic protein of the Cajal body, which is a conserved nuclear organelle that participates in many aspects of small ribonucleoprotein (RNP) biogenesis (*Gall et al., 1999*). Coilin is necessary for Cajal body formation and recruitment of spliced nucleoproteins (snRNPs) to modify protein complexes that guide RNA and motor neurons (SMN). When coilin is deleted, the remaining Cajal bodies lose contact with SMN complexes. Because SMN is considered a necessary condition for snRNPs assembly and internal circulation, the lack of interaction between SMN and coilin in the nucleus may lead to a decline in RNP assembly capacity, which may have down-stream effects on development and gametes, thus, affecting litter size. *Walker, Liping & Gregory (2009)* studied the effect of coilin removal on the overall viability and reproductive success of mice. The results showed that the number of oocytes that could fertilize was significantly reduced after *COIL* knockout, and the number of litters per foetus was low, showing obvious fertility and reproductive deficiencies (*Walker, Liping & Gregory, 2009*). *COIL* was found to be closely related to litter size. In this study, the association between genotype and litter size revealed that the CC genotype was significantly associated with litter size, and we verified that *COIL* is one of the candidate genes affecting the litter size in sheep.

## Relationship between *GUCY1A1* gene and sheep reproduction

Guanylate cyclase (GC) is an enzyme that converts guanylic-5′-triphosphate into cycloguangan-3′, 5′-monophosphate (*Pyriochou & Papapetropoulos, 2005*). As a membrane-binding molecule, GC exists in the membrane-bound and cytoplasmic forms. Soluble GC (sGCX), an isodimer enzyme composed of alpha and beta subunits, is the only receptor for nitric oxide (NO) as a biological messenger identified to date and is closely

involved in various signal transduction pathways (*Peter et al., 2010*). The sGCX collector (GUCY1A1) was found to be involved in hormone regulation, oxytocin signalling pathway, and reproductive capacity. In this study, we found that the mutation site of the *GUCY1A1* gene (g. 43266624C>T) was significantly correlated with litter size of Hetian sheep, thus, verifying the correctness of the sequencing results.

The results of KASP typing showed that three genes (*FSHR, GUCY1A1,* and *COIL*) were successfully typed. The genotypes of mutated loci correlated with high fecundity and average litter size of field sheep. The results showed that the genotypes of *FSHR* (g.75320741G>A) were extremely significantly correlated with high fecundity and average litter size of field sheep. *GUCY1A1* (g.43266624C>T) and *COIL* (g.7321466C>G) genes were significantly correlated with high fecundity and average litter size of field sheep. The results of this study confirm that mutations of these three genes caused changes in litter size of sheep and validated the SLAF sequencing technology.

Molecular marker screening is a major step in molecular breeding. SNP markers are third-generation molecular markers and are currently the mainstream molecular markers. KASP genotypes SNPs and indels at specific sites for precise genetic determination. Compared to other detection methods, KASP has higher throughput, is faster, more cost-effective, and produces more accurate results.

Breeding sheep with high fertility is important for mutton production. Molecular marker screening is used to improve early molecular selection. In this study, the KASP technology was used to verify three sites with significant or extremely significant effects on litter size, which could lead to breakthroughs in the development of sheep multi-lamb genes and markers, and improvement of the efficiency of molecular breeding of sheep with high fecundity. It can be used to improve other low-community groups and the efficiency of meat production in sheep. The study showed that SLAF-seq and KASP are cost-effective tools for selecting the desirable genotypes in sheep breeding programs.

## CONCLUSIONS

In this study, DNA of Chinese indigenous sheep from southern Xinjiang with different litter size fecundity were sequenced using the SLAF-seq technique. GWAS of SNPs that might cause differences in litter size was carried out. In total, 155 genes with significant mutations were obtained using SNP loci, gene annotation, and pathway analysis, of which 17 genes were related to reproductive traits. Seven candidate genes closely related to litter size and ovulation were identified through literature search and comprehensive analysis. The KASP technique was used to verify the role of the seven genes. Among the seven selected genes, only *FSHR* (GenBank ID: 443299, g. 75320741G>A site), *GUCY1A1* (GenBank ID: 101110000, g. 43266624C>T site), and *COIL* (GenID: 101123134, g. 7321466C>G site) loci were identified to be significant among different genotypes as molecular markers to detect the differences in litter size of sheep population from Xinjiang. This finding provides new information that explains the regulatory mechanism underlying sheep fecundity and identifies molecular markers for litter size traits. The sub-markers are of great significance for breeding of high-fecundity sheep breeds.

## ACKNOWLEDGEMENTS

We would like to thank all the participants of the study. We would also like to thank Editage for English language editing.

### Funding

This work was supported by the National Natural Science Foundation of China (No. U1603232), the National Key R&D Program of China (No. 2018YFD0502100), the Autonomous Region Key Research and Development Project (No. 2017B01005-2-1), and the Xinjiang Agricultural University Graduate Research and Innovation Project (No. XJAUGRI2017033). The funders had no role in study design, data collection and analysis, decision to publish, or preparation of the manuscript.

### Grant Disclosures

The following grant information was disclosed by the authors:
National Natural Science Foundation of China: U1603232.
National Key R&D Program of China: 2018YFD0502100.
Autonomous Region Key Research and Development Project: 2017B01005-2-1.
Xinjiang Agricultural University Graduate Research and Innovation Project: XJAU-GRI2017033.

### Competing Interests

The authors declare there are no competing interests.

### Author Contributions

- Haiyu Ma conceived and designed the experiments, performed the experiments, analyzed the data, prepared figures and/or tables, authored or reviewed drafts of the paper, approved the final draft.
- Chao Fang, Jueken Aniwashi, Yiming Sulaiman and Wujun Liu conceived and designed the experiments, contributed reagents/materials/analysis tools, authored or reviewed drafts of the paper, approved the final draft.
- Lingling Liu analyzed the data, prepared figures and/or tables, authored or reviewed drafts of the paper, approved the final draft.
- Qiong Wang performed the experiments, authored or reviewed drafts of the paper, approved the final draft.
- Kezierkailedi Abudilaheman conceived and designed the experiments, contributed reagents/materials/analysis tools, prepared figures and/or tables, approved the final draft.

### Ethics

The following information was supplied relating to ethical approvals (i.e., approving body and any reference numbers):

The University of Xinjiang Agricultural University approval to carry out the study within its facilities (Animal protocol number: 2017010).

## Data Availability

Raw data is available as a Supplemental File.

## Supplemental Information

Supplemental information for this article can be found online at http://dx.doi.org/10.7717/peerj.8079#supplemental-information.

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
