# Peer review of "Identification of novel genes associated with litter size of indigenous sheep population in Xinjiang, China using specific-locus amplified fragment sequencing technology"

_PeerJ, doi:10.7717/peerj.8079_

## Round 0.1 · original submission · Major Revisions

Dear Dr. Ma,

As the editor of your manuscript I advise you to follow of the comments of the three reviewers. Reviewer #2 points out that the methods section were not described with sufficient detail and that the data shared with the manuscript is not sufficient to replicate the experiment. In addition a more detailed description of the sheep populations is also essential. Both reviewers #1 and #3 felt that the discussion was insufficient, so please address this issue as well. Besides that please evaluate the minor changes suggested by all reviewers.

Reviewer 1 ·

Basic reporting

No comment.

Experimental design

No comment.

Validity of the findings

No comment.

Additional comments

1. Lines 92-93: The authors mention that “Hpy166II + EcoRV-HF enzyme was selected for enzymatic digestion (Jin et al., 2018)”. However, I could not find this reference! They need to explain the basis for selection of these enzymes.
2. Lines 94-96: At the same time, in order to evaluate the accuracy of the digestion strategy, Oryza sativa indica (http://rapdb. dna. affrc. go.JP/) was selected to evaluate the digestion efficiency (Li et al., 2009b).
Kindly explain these lines in context of this paper.
3. Line 141: Using EquCab2.0 sequence information of sheep genome……There seems to be some misunderstanding here.
EquCab2.0 is for Equus caballus (horse) and not sheep.
4. The authors have not discussed their results at all. They have just written about functions and previous studies regarding 3-4 genes.

Reviewer 2 ·

Basic reporting

Please see "General comments for the author" section.

Experimental design

Methods section were not described with sufficient detail. With the current format of the paper it is not possible to replicate the experiment. Please see "General comments for the author" for details.

Validity of the findings

The data shared with the manuscript is not sufficient to replicate the experiment. The SNP data provided is very limited and no phenotype/ trait information was shared.

Additional comments

The manuscript by Ma et al. describes a GWAS analysis resulting in the identification of SNPs which might be associated with litter size in indigenous sheep population in Xinjian, China. Litter size is economically the most important trait in lamb meat production and hence this manuscript dealt with a very important topic for sheep breeders. However, I’ve evidenced a number of issues and possible flaws that have to be addressed before being able to assess the validity and sound of the results obtained in this manuscript.

Major Compulsory Revisions:
Line 80 – 85: Needs to describe more about the sheep populations. Are they very similar breeds or are they very different breeds? As mentioned in the introduction (lines 61-62), different breeds possess different major genes affecting their litter size, please justify the reason doing GWAS combining two populations. What was the pedigree structure of the sheep populations? Are there any half sib family, etc? In my opinion it would be better to add MDS plot or PCA or neighbour network to better understand the population substructures (if any) between Hetian and Bashbay sheep papulation and within population variability.
Lines 107 – 111: This is not clear which SNP analysis are you refereeing at line 107. More description is needed for the GLM and MLM models. Please describe the MLM’s fixed effects and random effects. Why do you need to fit two different models? It was not clear how a Manhattan plot (Manhattan chart analysis) was used to analyse population structure? There was no Q-Q plot included in the manuscript. A Q-Q plot and reporting of inflation factor are needed to check whether or not their p-values were overestimated. No need to describe what programming language was used to create plots (Manhattan Diagram). “… and R was used to map.” (to map what?)
Lines 131 – 132: A description of the quality control filters is needed in Materials & Methods section.
Lines 135 – 139: Is there any correction for multiple testing? How have the authors reached their p-value threshold of 5.19e-7? With the analyses in their current form, I am not convinced that the authors can conclude that any, let alone 155 SNPs are
associated with litter size trait in Xinjiang sheep. Is there any explanation why in GLM the number of significant SNP is more that 2.5 times than that of MLM? Is there any overlap between SNPs found in GLM and MLM?

Minor Essential Revisions:
Some typos like “.,” at the end of line 77 needed to be fixed.
Line 41, 172, 257 and 272: is that C>T or G>A? As per description in lines 174-175 this should be G>A.
Line 165: Nine (sites/loci?) of the final three genes.
Line 171: Box plots might be better to understand than Table 2.
Line 207 – 208: Reference is needed for LOC101107541 and LOC101107119 genes.
Line 211 – 212: “.. FSHR gene may be one of the important genes affecting litter size
of Hetian sheep”. What about Bashbay sheep?
In Table 1 for position columns, is that the SNP position? The primer sequence information could be added in Appendix and in Table 1 this information could be replaced with a short description of the genes like function, chromosome, etc.
The legends for Figures and footnotes for tables needed to be expanded.

·

Basic reporting

no comment

Experimental design

no comment

Validity of the findings

no comment

Additional comments

Overview: DNA from the Chinese indigenous sheep from southern Xinjiang with different litter size fecundity were sequenced by the SLAF-seq technique. The work is well written and has clear objectives. It has interesting results and a product of application in the genetic improvement of sheep, the marker genes.

Introduction: It is necessary to be constructed a review of the distribution, prevalence and economic importance of the sheep production market

Material and Methods- Why 126 individuals were subjected to SLAF library construction and not 129?

Results - Most of the results are clear. Nevertheless, there are minor points to be clarified:
Table 1 - Are 5 genes cited or 4? The numerical values are not corresponding throughout the text. I suggest a complete review of the numbers throughout the text. Other forms of evaluation such as quantitative RT-PCR are important for validating marker genes to be used in animal breeding programs.

Discussion- This work is applied research. How would this study and its results impact the sheep breeding programs? It is important that this information is included in the final document. Also, it is necessary for a deeper discussion, citing an effective marker gene already used in sheep breeding.

---

## Round 0.2 · Minor Revisions

Dear Dr. Ma,

Although the manuscript has been improved from its previous version Reviewer #2 still raised minor issues, which I believe are essential to improve your manuscript. Please also re-examine the manuscript for typos and minor language issues.

Reviewer 2 ·

Basic reporting

No comment

Experimental design

No comment

Validity of the findings

No comment

Additional comments

Thanks to the authors to address my concerns raised in my previous review. I think the manuscript has improved after addressing the issues raised by all the reviewers. Just few bits and pieces:
In “peerj-36285-Manuscript(tracked_changes).docx”
Line 33: A total of 685,300 SNPs was identified by..
Line 113, 120, and 121: please add reference for PLINK, Admixture and SPAGeDi software
Line 229: Remove the word “concerns” from “They are only concerns about ovulation rate, low attention to litter size”
Line 268: In this study, it was found that the 75320741 sites of the FSHR gene was successfully classified into three genotypes, namely GG, GA, and AA, in the breeds of Xinjiang Tianyang.
Please double check the number. How a gene has so many sites?

---

## Round 0.3 · accepted · Accept

Dear Dr Ma,

Congratulations, after a few rounds of revisions I believe your manuscript has been largely improved.

Congratulations